# Feral Swine as Indirect Indicators of Environmental Anthrax Contamination and Potential Mechanical Vectors of Infectious Spores

**DOI:** 10.3390/pathogens12040622

**Published:** 2023-04-20

**Authors:** Rachel M. Maison, Maggie R. Priore, Vienna R. Brown, Michael J. Bodenchuk, Bradley R. Borlee, Richard A. Bowen, Angela M. Bosco-Lauth

**Affiliations:** 1Department of Biomedical Sciences, Colorado State University, Fort Collins, CO 80521, USA; rachel.maison@colostate.edu (R.M.M.); richard.bowen@colostate.edu (R.A.B.); 2Department of Microbiology, Immunology, and Pathology, Colorado State University, Fort Collins, CO 80521, USA; m.priore@colostate.edu (M.R.P.); brad.borlee@colostate.edu (B.R.B.); 3U.S. Department of Agriculture, Animal and Plant Health Inspection Service, Wildlife Services, National Wildlife Research Center, Fort Collins, CO 80521, USA; vienna.r.brown@usda.gov; 4U.S. Department of Agriculture, Animal and Plant Health Inspection Service, Wildlife Services, San Antonio, TX 78269, USA; michael.j.bodenchuk@usda.gov

**Keywords:** anthrax, *Bacillus anthracis*, feral swine, *Sus scrofa*, sentinel, mechanical vectors, spores, experimental inoculation, ELISA, surveillance

## Abstract

Anthrax is a disease that affects livestock, wildlife, and humans worldwide; however, its relative impacts on these populations remain underappreciated. Feral swine (*Sus scrofa*) are relatively resistant to developing anthrax, and past serosurveys have alluded to their utility as sentinels, yet empirical data to support this are lacking. Moreover, whether feral swine may assist in the dissemination of infectious spores is unknown. To address these knowledge gaps, we intranasally inoculated 15 feral swine with varying quantities of *Bacillus anthracis* Sterne 34F2 spores and measured the seroconversion and bacterial shedding over time. The animals also were inoculated either one or three times. The sera were evaluated by enzyme-linked immunosorbent assay (ELISA) for antibodies against *B. anthracis*, and nasal swabs were cultured to detect bacterial shedding from the nasal passages. We report that the feral swine developed antibody responses to *B. anthracis* and that the strength of the response correlated with the inoculum dose and the number of exposure events experienced. Isolation of viable bacteria from the nasal passages of the animals throughout the study period suggests that feral swine may assist in the spread of infectious spores on the landscape and have implications for the identification of environments contaminated with *B. anthracis* as well as the exposure risk to more susceptible hosts.

## 1. Introduction

Anthrax is a zoonosis caused by the spore-forming bacterium *Bacillus anthracis*. It affects wild and domestic mammals worldwide. Though most mammals are susceptible to infection, the severity of the disease differs largely between taxa, with ruminants often dying within a few days of exposure, while carnivores, omnivores and scavengers frequently survive and can develop measurable immunity to infection [1,2,3,4,5]. Exposure in animals most frequently occurs after ingesting *B. anthracis* spores from contaminated soil, water, or vegetation, although in some environments biting flies as well as other necrophilic and hemophagic arthropods can also transmit the bacteria and result in infection [6,7,8,9]. Inhalation of aerosolized spores is also a possible route of exposure; however, limited evidence to date has supported this as a significant mode of transmission for ruminants in natural settings [10,11]. Spillover to humans most often occurs when contaminated animals or animal products are directly handled or consumed [12].

Once inside a viable host, *B. anthracis* changes into its vegetative form and pathogenesis is established with the presence of toxins and capsules, which are two virulence factors encoded on the bacterium’s pXO1 and pXO2 plasmids, respectively [13]. Lethal factor (LF) and edema factor (EF) proteins encoded on the toxin plasmid enter host cells through the mediation of the cell receptor-binding protein protective antigen (PA) encoded on the toxin plasmid, which also enables the bacteria to produce LF and EF toxins. These toxins are ultimately responsible for the subsequent death of the host [13,14,15]. After host death, vegetative bacilli exposed to atmospheric oxygen sporulate and contaminate the surrounding environment, where they may either stay localized or be further distributed with the assistance of arthropod or vertebrate vectors, or a variety of abiotic factors, before infection of another suitable host takes place [9,16,17]. Once sporulated, *B. anthracis* is highly resistant to environmental degradation and can survive upwards of 100 years in appropriate conditions, making environmental decontamination a difficult disease management option [18,19,20].

Despite being considered a disease of antiquity, the relative impacts of anthrax on susceptible animal populations remain largely undescribed [4]. This is particularly true for wild ruminants, as these populations are highly susceptible to fatal anthrax and are not as easily observed for clinical disease as domestic animals, making it likely that the true incidence is underestimated [21]. Compounding the unknown incidence in wildlife, ongoing surveillance in highly susceptible domestic populations such as livestock is largely passive or opportunistic and involves observing animals for clinical signs and quickly reporting and disposing of any suspicious carcasses [22,23]. Preventative management, where existing, is primarily vaccine-based, although vaccination is not a requirement for livestock owners located in highly endemic regions and is instead often used as a reactionary measure to control disease spread after an outbreak has begun rather than as a preventative action [23,24]. Moreover, the current vaccination procedure used for livestock requires animals to be given an annual subcutaneous injection, which is not a viable strategy for managing disease in wild or free-ranging ruminant populations. Thus, in highly endemic regions of the United States such as western Texas, where anthrax enzootics frequently occur, the disease is further perpetuated on the landscape and remains a problem in wild ruminants such as the white-tailed deer (*Odocoileus virginianus*) [9,25].

Given that not all animal species are equally impacted by anthrax, it has been suggested that more resistant species may be used as sentinels for either outbreak management or ongoing surveillance efforts, as they often exhibit a high prevalence of antibodies against the bacterium after exposure [4,5]. Furthermore, surveys of scavengers such as vultures have recovered viable spores from both feces and the surfaces of feathers after they have scavenged on infected carcasses, suggesting that anthrax may be disseminated significant distances with the assistance of some vertebrate vectors [12,26,27]. Members of the Suidae family, which include the Eurasian wild boar, domestic swine, and invasive feral swine in the United States (all *Sus scrofa*), are also resistant to anthrax disease [12,28]. Wild or free-ranging suids are of particular interest for their potential biosentinel utility, as past serological surveys have detected anti-anthrax antibodies in wild-caught individuals, and in the case of feral swine, are ubiquitously distributed in some regions [29,30,31]. Additionally, pigs are known for their close associations with soil through their distinct rooting and wallowing behaviors, and they have been hypothesized to be exposed to anthrax spores in contaminated soils at high rates compared to other species [30,32]. Despite these hypotheses, pigs, and particularly feral swine, have yet to be examined for their role in anthrax epidemiology beyond simple exposure to the bacterium, particularly if they may be vectors themselves and assist in the spread of infectious spores on the landscape. To address these knowledge gaps, we assessed a group of feral swine for their ability to seroconvert and shed viable *B. anthracis* after intranasal exposure to known levels of the bacteria over time.

## 2. Materials and Methods

### 2.1. Animals

Wild-caught juvenile feral swine of mixed sex were captured in Burnet County, Texas, before being transported to Colorado State University. Feral swine sounders were trapped by the United States Department of Agriculture, Animal and Plant Health Inspection Service, Wildlife Services (USDA) personnel in Texas using baited corral traps from 25 to 27 April 2021 during routine feral swine damage management operations. Juvenile swine were preferentially selected from the sounders to minimize the chances of previous exposure to anthrax. Juveniles were defined as less than 2 months of age, and age was estimated based on the absence of incisors and permanent canines on the lower jaw [33]. Upon capture, individuals were selected based on these criteria and loaded onto a USDA-designated trailer, wherein they were bed on straw, provided with water and food ad libitum and transported overnight to Colorado. The animals were then group-housed in an Animal Biosafety Level 3 (ABSL-3) facility at Colorado State University, which comprised two rooms (12 ft × 18 ft) with natural lighting and controlled climate. The swine had access to food and water ad libitum throughout the acclimation and study period.

The animals were allowed to acclimate for six days before being given a thorough physical examination, where an initial blood sample was taken and individuals were screened externally for ectoparasites, sexed, and implanted with thermally sensitive microchips (Lifechips, Destron-Fearing, St. Paul, MN, USA) for identification and temperature measurement. Due to the significant numbers of ectoparasites being observed at this time, a thin application of deltamethrin insecticide (Delta Dust, Bayer, Cary NC, USA) was also applied to the coat of each pig during the initial exam. Blood samples were collected and then examined by enzyme-linked immunosorbent assay (ELISA), as described below, to screen for previous or recent exposure to *B. anthracis*. Fifteen confirmed seronegative pigs were randomly assigned to one of five study groups (*n* = 3 per group), as summarized in Table 1. Depending on the group assignment, the pigs were challenged with *B. anthracis* Sterne spores with either 10^4^ CFU/mL or 10^7^ CFU/mL, and they experienced either one or three inoculations. Three pigs were designated as positive serological controls and were given a single subcutaneous dose of the standard livestock vaccine, equating to 1 mL and 10^4^ CFU Sterne spores. The pigs were housed such that individuals experiencing equal numbers of exposures were housed together (Table 1). This was done to minimize cross-contamination and ensure that individuals were only exposed to the inoculum level desired. All the methods and uses of animals described here were performed with Colorado State University’s Institutional Animal Care and Use Committee approval (Protocol #1535).

### 2.2. Spore Preparation

One 10 mL vial of commercial Anthrax Spore Vaccine (Serial #471, Vet License No. 188) containing *B. anthracis* Sterne 34F2 was acquired from Colorado Serum Company. To prepare a new working stock, 40–100 µL of the vaccine was streaked onto a chocolate agar plate using a sterile inoculating loop. After overnight incubation at 37 °C, the resulting vegetative *B. anthracis* Sterne was then sporulated as described previously to prepare a stock of inoculum [34]. Briefly, one colony from the overnight plate culture was inoculated into 10 mL of pre-warmed 2X SG medium [35] and incubated on a shaker at 37 °C. A 2.5 mL sample of this preculture was then added to 100 mL of pre-warmed 2X SG medium and shaken at 37 °C for 24 h. After the 24 h incubation, 400 mL of double-distilled water was added to the culture, and incubation with shaking resumed for an additional 40 h. After 40 h, the culture was transferred into three sterile Oakridge tubes and centrifuged in a Sorvall Superspeed centrifuge with an SS-34 rotor at 4–10 °C and 4000 rpm for 20 min. The resulting pellets were then washed once with sterile PBS (50 mL per tube) and centrifuged again for 20 min at 4000 rpm before being combined and resuspended in 40 mL sterile PBS. The final spore suspension was then heated to 68 °C for 40 min to kill any residual vegetative cells present in the solution. The final spore solution was then transferred into 1 mL aliquots and stored at −80 °C. The stock titer was determined to be approximately 10^6.9^ CFU/mL after serial dilution onto Polymyxin B-Lysozyme-EDTA-Thallous acetate agar (PLET) plates (Sigma, #55678, St. Louis, MO, USA).

### 2.3. Animal Challenge

The prepared aliquots of *B. anthracis* Sterne spores were thawed immediately prior to the animal challenge and diluted in sterile PBS to concentrations of either 10^4^ CFU/mL or 10^7^ CFU/mL. Notably, since the estimated titer of the bacterial stock was slightly lower than that required for the high-dose groups, the inoculum volume for the high-dose groups was raised slightly from 1 mL to 1.2 mL to allow for the desired dose to be administered. The diluted *B. anthracis* Sterne spores were then administered to the nares of 12 individuals using a pipette (500 µL per nare for low-dose groups and 600 µL per nare for the high-dose groups) under manual restraint. The three individuals serving as positive controls were given 1 mL of the Anthrax Spore Vaccine subcutaneously, as recommended by the manufacturer. For those groups experiencing three exposures, the inoculum was prepped twice more as described above and administered once every other day until three total doses were administered.

### 2.4. Sampling and Clinical Observation

We evaluated all the feral swine for seroconversion as well as bacterial shedding in the nasal passages following single and repeat exposures to *B. anthracis* Sterne. Both blood and nasal swab samples were collected from all the individuals at 14, 28, 42, and 56 days post-inoculation (DPI); however, the animals that experienced one intranasal exposure event were only observed to 28 DPI before being euthanized. Removing these individuals from the study at 28 DPI was determined based on the average time the vertebrate adaptive immune system takes to mount a complete response to an antigen that is introduced to the system for the first time being between 1 and 2 weeks [36]. Blood (3–5 mL into serum separator tubes) was collected from the anterior vena cava vein under manual restraint, and serum was extracted within two hours of collection. All the sera were passed through a 0.22 µm filter before being taken out of the ABSL-3 and stored at −80 °C pending serological analysis. Nasal swabs were collected by rotating a standard polyester swab (Puritan, #25-806) inside the nasal passage for five seconds while under manual restraint. The swabs were then broken off into tubes containing 1 mL of phosphate-buffered saline (PBS) with 20% glycerol and stored at −80 °C pending cultivation, DNA extraction from suspicious colonies, and confirmation by PCR.

All the animals were otherwise observed daily, which included an assessment of temperament and the presence or absence of clinical signs of disease, including but not limited to ocular and nasal discharge, ptyalism, coughing/sneezing, dyspnea, diarrhea, lethargy, anorexia, and moribund status. Body temperatures were also observed during the days of sample collection when the animals were being handled.

### 2.5. Serology

We evaluated all the feral swine sera for the presence of anti-anthrax antibodies by ELISA, as described previously [29]. Briefly, high-binding polystyrene microtiter plates were coated with recombinant protective antigen (rPA) from *B. anthracis* (ATCC, #NR-3780, Manassas, VA, USA) at a concentration of 5 µg/mL per well and allowed to incubate overnight at 4 °C. After overnight incubation, the plates were washed five times with 300 µL of washing buffer (0.05% Tween 20 in PBS) and blocked for 1.5 h with 10% skim milk in PBS, followed by additional washing. Feral swine serum diluted to 1:100 in 1% skim milk in PBS was then added in triplicate to the plate wells and incubated with shaking at room temperature. After additional washing, 100 µL of protein A/G-horseradish peroxidase (HRP) (Thermo-Scientific, #32490, Rockford, IL, USA) was added to the wells and incubated for 30 min at room temperature. After one final washing step, 150 µL of ABTS substrate solution (Thermo-Scientific, #37615, Rockford, IL, USA) was added and allowed to develop for 15 min. The reaction was stopped with the addition of 100 µL of 1% sodium dodecyl sulphate, and the sample absorbance was measured at 25 °C and 405 nm. Samples were considered positive for the rPA antibody if their mean absorbance were above three times the standard deviation of the mean of the negative control sera, equating to 0.15 absorbance units. The known negative and positive control sera utilized for each assay were obtained from a male domestic goat (*Capra aegagrus hircus*) before and after vaccination with Anthrax Vaccine Absorbed (BioThrax^TM^, #NR-2642, Lansing, MI, USA), also as described previously [29].

### 2.6. Nasal Swab Processing

All the nasal swab samples were thawed completely before being placed in a 65 °C water bath for 30 min to preselect for endospore-forming bacteria [37]. The tubes were then vortexed briefly, and 100 µL of swab media was plated onto the PLET plates. The plates were then incubated for 18–24 h before being observed for colonies with characteristic *B. anthracis* morphology [38].

### 2.7. DNA Extraction and PCR

DNA extraction was performed by suspending one colony in 100 µL of TE buffer (10 mM Tris-HCl [pH 7.6], 1 mM EDTA [pH 8]) and heating at 95 °C for 20 min. Multiplex PCR was performed using Thermo Fisher Scientific Platinum Multiplex PCR Master Mix (Cat. #4454268) and the *cap* and *pag* primers, as described previously [39]. A total volume of 25 µL reactions containing 0.2 µM of *cap* and 0.05 µM of *pag* primer sets and 1 µL of template DNA was used. The PCR program consisted of an initial denaturation at 94 °C for 1 min, followed by 35 cycles each of denaturation at 94 °C for 30 s, annealing at 58 °C for 2 min, and extension at 72 °C for 2 min. The program was completed with a final extension at 72 °C for 10 min. The amplified products as well as a 1 kilobase ladder (Invitrogen, #10787018, Waltham, MA, USA) were then loaded onto 1% agarose gel stained with ethidium bromide and electrophoresed for 1.5 h at 100 V. Each PCR assay contained three control strains, each exhibiting unique pXO1/pXO2 profiles for quality assurance, as detailed in Table 2. Products were considered positive for *B. anthracis* Sterne if they exhibited a band profile consistent with the plasmid content of *B. anthracis* Sterne, constituting a band of the appropriate size for *pag* (364 bp) and no band presence for *cap* (578 bp).

## 3. Results

None of the feral swine displayed any outward signs of clinical disease following exposure to *B. anthracis* Sterne intranasally or subcutaneously, and they all maintained stable body temperatures throughout the study period. Moreover, none of the study animals displayed any abnormal behavior(s) compared to the acclimation period.

The serological results, as determined by ELISA, are summarized by group in Figure 1. All the study groups, except for those experiencing the lowest exposure doses of 10^4^ CFU/mL administered once intranasally, seroconverted and displayed anti-PA antibodies, which are depicted as absorbance values above the assay’s cutoff of 0.15 absorbance units (Figure 1). All the other intranasally exposed pigs that did seroconvert exhibited antibody levels and seroconversion rates that positively correlated with their level of exposure. The pigs inoculated with 10^4^ CFU/mL exhibited less of a response than those inoculated with 10^7^ CFU/mL, and the animals experiencing a single exposure were lower than those experiencing three exposures. The pigs in the positive control group who were given the standard livestock vaccine of 10^4^ CFU/mL subcutaneously displayed the highest absorbance values throughout the study period and seroconverted the fastest. The antibody levels in most pigs peaked at 28 DPI, and in most individuals that were kept beyond 28 DPI, started to decline at similar rates between 42 and 56 DPI. The two exceptions were pigs in the intranasal 10^4^ CFU/mL group exposed three times and those in the 10^7^ single-exposure group, who both peaked in absorbance values at 42 and 28 DPI, respectively.

The results of the nasal swab culture and subsequent colony PCR confirmation performed on all the intranasally exposed pigs are depicted in Table 3. *B. anthracis* Sterne was successfully isolated from the nasal passages of the majority (10/12) of pigs at one or more time points post-inoculation. For the single exposed individuals kept only until 28 DPI, three of the six had detectable *B. anthracis* Sterne colonies out to 28 DPI, one in the 10^4^ and two in the 10^7^ CFU/mL exposure groups; the other three pigs were negative at the time of euthanasia. Regarding the groups exposed multiple times, all six pigs had positive cultures on 14 and 28 DPI. The majority (5/6) of pigs that experienced multiple exposures were positive on 42 DPI, and 2/6 were positive on 56 DPI, one exposed to 10^4^ and one to 10^7^ CFU/mL. While most pigs that displayed positive cultures post-exposure were positive on 14 DPI and either stayed positive or became negative at the next time point, three individuals in the single-exposure groups (#7634, 7641, and 7636) were negative during this initial sampling point and instead became positive on 28 DPI. The relative bacterial loads within the swab samples also were correlated with the exposure level, as more Sterne colonies were observed from the samples taken from individuals exposed to higher inoculum doses as well as those exposed multiple times (Table 3). For most swine, the bacterial load from the nasal swabs also declined substantially over time, with individuals experiencing between 4- and 100-fold reductions in the colony counts between time points. Exceptions to this trend were three pigs in the single-exposure groups, who exhibited Sterne on the 14 DPI swab cultures but either 10 CFU/mL (#7634 and 7641) or 20 CFU/mL (#7636) on 28 DPI. Additionally, one individual (#3911) in the low-dose, multiple-exposure group experienced a slight increase in the bacterial load between 42 and 56 DPI, with 10 CFU/mL detected on 42 DPI and 30 CFU/mL on 56 DPI.

## 4. Discussion

After exposure to varying quantities of *B. anthracis* Sterne spores, the feral swine seroconverted and displayed measurable antibodies against anthrax PA. Additionally, the feral swine appeared to be competent hosts and possible vectors of *B. anthracis* spores after intranasal exposure, as viable bacteria were isolated from the nasal passages of most individuals weeks after their initial or final intranasal exposure.

While we observed our study pigs daily for clinical signs of disease, as well as observing the body temperatures at each sampling time point, we detected no abnormalities throughout the study period and no adverse responses after either intranasal or subcutaneous exposure to *B. anthracis* Sterne. This was not surprising given that the Sterne strain is an avirulent strain that is routinely used to vaccinate livestock with little risk [43,44]. Additionally, members of the Suidae family are thought to be relatively resistant to developing anthrax after exposure to fully virulent *B. anthracis*, suggesting that any clinical presentation of the disease would be a rare occurrence.

The strength of the antibody response observed in our study pigs was positively correlated with both the dose of the inoculum given and the number of exposures individuals experienced over time. The seroconversion observed after exposure to known quantities of *B. anthracis* was consistent with the serology observed in prior field surveys, where varying levels of anti-PA antibodies were detected in wild-caught swine with no known exposure status [29,30]. This supports past hypotheses postulating that swine may be indirect indicators of anthrax contamination. Of note, the only study group that did not seroconvert was the 10^4^ CFU/mL single-exposure group. This suggests that multiple exposure events or relatively high environmental contamination may be required for pigs to be useful as sentinels in regions with anthrax and where feral swine removal or invasive species management is taking place. Furthermore, the fact that most of our study pigs were exposed to *B. anthracis* Sterne intranasally and still seroconverted at levels detectable by ELISA validates that inhalational exposure is a legitimate exposure route for swine. The intranasal inoculation of our study pigs was meant to simulate an exposure event from pigs engaging in natural rooting behaviors and inadvertently inhaling infectious materials from their environment. Currently, inhalational exposure is believed to be the least common route of infection in livestock and wildlife, as anthrax spores tend to cluster together with surrounding organic materials in soil and are not considered easily aerosolized [45]. It is additionally thought that susceptible ruminants are most likely exposed to anthrax via the ingestion of spores. This has been proposed for grazing species such as bovids, as their grazing habits involve ripping grass with their tongues, an action likely facilitating the ingestion of soil, the main substrate for *Bacillus* spores [46]. For browsing ruminants, especially those whose ranges overlap with necrophilic flies, the ingestion of spores from browse is thought to be enabled by the vectoring of spores from infected carcasses to the leaves of vegetation by flies [6,7,8,9]. Exposure in resistant carnivorous and omnivorous species is also thought to be from the ingestion of spores, mainly from feeding on contaminated animals [12]. Unlike grazing and browsing herbivores, pigs interact with soil profiles very differently, most often obtaining food by utilizing their muzzle to upturn substrate and their sensitive sense of smell to locate suitable roots or grubs [32,47]. This behavior likely facilitates the aerosolization of infectious materials better than the grazing activities of other species, and it may be a more likely route of exposure for swine in natural settings. Like predators and scavengers, feral swine also are known to predate on wildlife as well as scavenge upon carcasses; therefore, they may also experience gastrointestinal anthrax exposure in some instances [30]. Regarding how these exposure routes might influence the resulting serosurveillance efforts, follow-up studies will be needed to better elucidate any differences in the seroconversion of pigs experiencing gastrointestinal versus inhalational anthrax exposure.

In addition to the marked seroconversion, the pigs that experienced intranasal exposure to *B. anthracis* Sterne also retained observable quantities of inoculum within their nasal passages for varying periods of time after inoculation. The bacterial culture and subsequent PCR confirmation of the colonies isolated from the nasal swab samples revealed that for some pigs, *B. anthracis* Sterne stayed present at detectable levels weeks after an exposure event occurred. This was particularly true for pigs exposed multiple times, as *B. anthracis* Sterne was recoverable in some individuals for as long as 42 and 56 days following primary inoculation. As pigs have a natural propensity to root in soil, this suggests that in environments contaminated with *B. anthracis*, feral swine may serve as mechanical vectors by inadvertently inhaling soil particles containing spores into the nasal passages, where they may then adhere to the nasal mucosa and persist for long periods of time. Notably, for three pigs in the single-exposure groups, *B. anthracis* Sterne was not detected in the nasal swabs at 14 DPI but was at 28 DPI. This pattern, as well as the continued recovery of bacteria from pigs inoculated multiple times, may be explained either by the persistence of the inoculum or by the replication of vegetative cells that have colonized the nasal epithelium. Given the lack of clinical abnormalities in any of the pigs, as well as the avirulent nature of the Sterne strain, the replication and preservation of vegetative cells in the nasal passages are unlikely. Furthermore, the decreasing colony counts observed over time from the nasal swab cultures of the intranasally inoculated pigs do not support continued bacterial replication in the nasal passages. This suggests that *B. anthracis* Sterne spores were recovered from the nasal passages, and therefore, the persistence of the inoculum rather than the establishment of bacterial infection in the epithelium. Indeed, it has been suggested by others that species resistant to anthrax predating or scavenging on infected animals can act as carriers by dispersing spores via dried viscera adhered to fur or feathers, or in feces after ingestion of contaminated materials [7,20,27]. Mechanical vectoring of viable spores has also been demonstrated for some arthropods, including various necrophagous flies as well as some mosquitos [7,48,49]. Additional movement of anthrax spores, especially in regions where feral swine are present, could therefore have significant implications for the exposure of more susceptible species, and consequently, how long an outbreak may last. This may especially be true in instances where susceptible wildlife and feral swine are in competition for the same food sources, as has been proposed between feral pigs and white-tailed deer in regions of Texas during times of drought [50]. White-tailed deer are also known to visit feral swine wallows, and this additional source of interaction likely has further implications for the transmission of diseases such as anthrax [51].

This experimental infection study had limitations. Chief among them was the utilization of an avirulent strain of *B. anthracis* for inoculation rather than a wild-type or fully virulent strain. Exposure to *B. anthracis* Sterne in a species already relatively resistant to developing anthrax may have resulted in the underestimation of the subsequent serological response of each study group, as the Sterne strain does not possess the capsule-encoding pXO2 plasmid and thus does not interact with the host immune system in the same way fully virulent pXO1- and pXO2-expressing strains do [15,43]. The Sterne strain does express the toxin-encoding pXO1 plasmid, which includes the cell receptor-binding protein PA. It has been demonstrated previously that the bulk of the humoral immune response generated to anthrax is against PA and that higher anti-PA antibody titers are correlated with increased levels of protection against the disease [52]. This is further supported by resistant wildlife species in regions endemic for anthrax displaying measurable antibodies against PA [1,2,3,4,5], and it suggests that despite challenge with a non-virulent strain, the antibodies we observed in our feral swine very likely mimic those that are observed in natural systems containing wild-type anthrax. However, comparing antibody levels using a quantitative ELISA platform may be necessary to further elucidate if differences exist in the serology of swine exposed to different strains of *B. anthracis*. Lastly, while the recovery of viable spores from the nasal passages after intranasal inoculation suggests the potential for mechanical transmission, we did not sample any of the substrates or surfaces within our animal rooms and thus cannot confirm that spores were then deposited elsewhere and available for other individuals to then be exposed. Additional experiments are necessary to verify spore deposition onto surfaces or substrates utilized by feral swine after intranasal exposure, and particularly if enough infectious material is deposited to then infect other individuals. Furthermore, while we also collected nasal swab samples from subcutaneously exposed pigs to assess the transmission of Sterne from intranasally inoculated pigs housed in the same room, the contamination of the swab samples collected on 0 DPI with inoculum confounded any conclusions to be drawn from samples collected at subsequent time points from those individuals. In other words, subcutaneously inoculated individuals appeared positive for *B. anthracis* Sterne within their nasal passages on the same day that they were inoculated, suggesting that cross-contamination occurred either within the nasal passages of those individuals or on the swab itself after or during sampling. Any positive swab samples appearing at subsequent time points, therefore, could not be interpreted as being the result of mechanical transfer between pigs. Further studies are, therefore, required to confirm whether spores may be transferred between the nasal passages of exposed and naïve swine.

Anthrax is largely considered a disease of antiquity, yet it remains a significant problem for domestic livestock and wild ruminants worldwide. Moreover, there remain significant knowledge gaps regarding *B. anthracis* environmental distribution and ecology, including the relative roles that resistant host species may play in bacterial dissemination [20]. Feral swine are a species resistant to anthrax and known for their regular interactions with soil, the primary substrate for infectious *B. anthracis*, although they have been overlooked for their role in anthrax ecology. We report that feral swine intranasally exposed to *B. anthracis* Sterne may serve as competent biosentinels based on seroconversion after exposure to known levels of bacteria. Measuring the antibody status of feral pigs may thus be used as a more targeted surveillance tool for determining whether an area poses an exposure risk for susceptible livestock or wildlife. In some cases, this information may assist livestock owners or wildlife managers in determining appropriate actions to mitigate risk or be used to support the use of livestock vaccination [4,53]. We additionally report that feral swine exposed intranasally to *B. anthracis* Sterne may serve as mechanical vectors for infectious spores based on carriage in the nasal passages weeks after a known exposure occurred. Management of feral swine in endemic and emerging anthrax regions may, therefore, not only include serosurveillance as a tool for determining anthrax contamination status, but also could incorporate removing individuals whose movements may be from regions experiencing outbreaks to avoid mechanical transmission to more susceptible species as well as humans.

## Figures and Tables

**Figure 1 pathogens-12-00622-f001:**
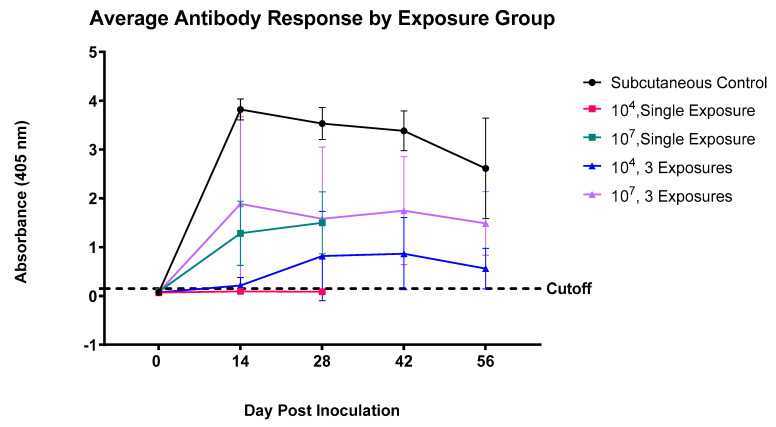
Average antibody response in feral swine exposed to various levels of *B. anthracis* Sterne 342 spores over time ± standard deviation for each group. “Subcutaneous Control” refers to swine that were given the standard livestock vaccine for *B. anthracis*, equating to 1 mL of 10^4^ CFU/mL of *B. anthracis* Sterne administered once subcutaneously. The dotted cutoff line represents the assay cutoff between the seropositive and seronegative animals (e.g., +3 SD above the mean of the negative control sera), equal to 0.15 absorbance units.

**Table 1 pathogens-12-00622-t001:** Study group assignments for juvenile feral swine exposed to varying amounts of *Bacillus anthracis* Sterne spores.

Animal ID	Sex	Times Exposed	Inoculum Dose (CFU/mL)	Route of Inoculation ^1^	Animal Room
7634	F	1	10^4^	i.n	1
7635	F	1	10^4^	i.n	1
7637	F	1	10^4^	i.n	1
7618	F	1	10^7^	i.n	1
7641	F	1	10^7^	i.n	1
7636	M	1	10^7^	i.n	1
7619	F	1	10^4^	s.c	1
7621	F	1	10^4^	s.c	1
7633	M	1	10^4^	s.c	1
3915	F	3	10^4^	i.n	2
3911	M	3	10^4^	i.n	2
3920	M	3	10^4^	i.n	2
3913	F	3	10^7^	i.n	2
3912	M	3	10^7^	i.n	2
3919	M	3	10^7^	i.n	2

^1^ i.n, intranasal; s.c, subcutaneous.

**Table 2 pathogens-12-00622-t002:** Plasmid characteristics and associated references for the bacterial strains used as controls for the multiplex PCR.

Species	Strain [Reference]	Plasmid Content	
		*pag*	*cap*
*Bacillus cereus*	UW85 [40]	−	−
*Bacillus anthracis*	Sterne 34F2 [41]	+	−
*Bacillus anthracis*	1075.4 [42]	+	+

**Table 3 pathogens-12-00622-t003:** Results of the nasal swab culture by study group (intranasally exposed swine).

Animal ID	Inoculum Dose (CFU/mL)	Times Exposed	Nasal Swab Culture ^1^ +/−, (CFU/mL)			
			D14	D28	D42	D56
7634	10^4^	1	−	+ (10)	NA	NA
7635	10^4^	1	−	−	NA	NA
7637	10^4^	1	−	−	NA	NA
7618	10^7^	1	+ (80)	−	NA	NA
7641	10^7^	1	−	+ (10)	NA	NA
7636	10^7^	1	−	+ (20)	NA	NA
3915	10^4^	3	+ (220)	+ (50)	+ (50)	−
3911	10^4^	3	+ (TN)	+ (60)	+ (10)	+ (30)
3920	10^4^	3	+ (390)	+ (110)	−	−
3913	10^7^	3	+ (670)	+ (180)	+ (20)	−
3912	10^7^	3	+ (270)	+ (20)	+ (80)	−
3919	10^7^	3	+ (110)	+ (160)	+ (50)	+ (10)

^1^ “+” indicates *B. anthracis* Sterne was detected after the nasal swab culture and confirmed by PCR. “−” indicates no detectable *B. anthracis* Sterne was isolated. NA: not applicable. Colony-forming units of *B. anthracis* Sterne detected per ml of swab media are indicated in parentheses. TN: too numerous to estimate.

## Data Availability

All relevant data are contained within this article.

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
