# Peer review of "Feral Swine as Indirect Indicators of Environmental Anthrax Contamination and Potential Mechanical Vectors of Infectious Spores"

_pathogens, 2023, doi:10.3390/pathogens12040622_

Round 1
Reviewer 1 Report
In the study « Feral swine as indirect indicators of environmental anthrax contamination and potential mechanical vectors of infectious spores», the authors proposed feral swines as biosentinels and as possible vectors of infectious spores. They exposed captured wild feral swine to different doses of the Sterne strain, one or three times. They evaluated the response of this resistant animal to anthrax with the humoral response developed (anti-PA antibodies by ELISA) and by the presence of bacteria in nasal swab. This is a substantial work with wild animals, very pleasant to read. It highlights the importance of monitoring and controlling the movements of anthrax-resistant animals. Yet, their resistance to anthrax makes feral swines good biosentinels but also possible vectors of the disease. The abstract is clear. The introduction is also very clear, fluid and correctly presents the subject. I am only surprised by the role of PA, which activates the bacteria to produce LF and EF toxins (line 52). Are you sure it is PA? Isn’t it rather atxA, carried on pXO1 ? Or else, please give the reference of this role of PA. The Materials and Methods part is very well described, with all the information necessary for a good understanding of the experiments. However, why animals that experienced one i.n exposure event were euthanized after 28 DPI? Can you explain it in this part? Because the level of antibody response is close to that observed with 107 spores, 3 exposures. The presentation of the results is also limpid and concise. Yet, you can add a table with the absorbance values at each time for each animal. This will give an interesting idea of the response variability with wild animals. The discussion is a bit long but it comes back well on the results. The different parts are interesting, especially the part that justifies the use of the Sterne strain. To reinforce the next use of this strain and to better compare the seroconversion following intranasal infection versus gastrointestinal infection, you should plan a quantitative ELISA for anti-PA antibodies. Lines 348-350, thank you for reviewing the sentence, a verb is probably missing. Lines 383-388, for subcutaneously pigs, I don't understand why samples collected at 0 DPI and contaminated at that time interfere with samples collected later. For feral swines infected via intranasal route, you only present the results at D14, D28, D42 and D56. Why not for the other pigs?
Reviewer 2 Report
The authors wanted to analyze the potential of feral swine to disseminate spores of B. anthracis. One could assume that these relatively resistant animals might contribute to the spread of spores due to their close association with soil through their rooting and wallowing behaviors. As empirical data were lacking, the authors wanted to fill this knowledge gap. Although they could show that swine immunized with the Sterne vaccine strain developed antibodies and contained viable intranasal bacteria, their real impact in spreading anthrax in natural settings remains speculative. In addition, despite of the undoubted effort to perform a study with 15 feral swine, there are some gaps in the study design and evaluation which unfortunately cannot be improved because the experiments cannot be repeated. This concerns especially the results of nasal swab culture, as indicated below.
Major comment:
Line 208/Chapter 2.6/Table 3/Lines 341-350:
The nasal swabs were heated at 65°C to select for spores. This procedure prevents detection of vegetative bacteria of B. anthracis which might also be present. It would have been interesting to know if spores were able to germinate and persist as vegetative cells in the nasal epithelium. The authors assume that “replication and preservation of vegetative cells in the nasal passage is unlikely” (line 347), but the presence of vegetative cells could have been tested by comparing heated and untreated aliquots of the nasal swabs. PLET agar is very selective for B. anthracis and the nasal flora should be largely inhibited even without heating.
In addition, it would be important to know not only if culture was possible or not (as indicated by +/- in Table 3), but also the approximate bacterial load. For example, in two animals inoculated once with 107 spores, no bacteria were cultivated at D14, but were detected at D28 (this also applies for one animal inoculated with 104 spores), whereas from one animal, bacteria were cultured at D14, but not at D28. If the number of bacteria is very low, there might be a statistic effect of “catching” one bacterium or not while swabbing the nose. However, if the number of bacteria was relatively high at D28 but no bacteria were detected at D14, this might indicate replication. If the number of CFU is still available (and was not shown), it should be presented in the manuscript. It would also be interesting to see a decrease in the number of bacteria over time in the groups inoculated three times. Knowing the number of bacteria present in the nasal passage would also help to estimate the risk of shedding.
Minor comments:
Line 38: The beginning of the sentence (A soil-dwelling bacterium...) should be revised, the wording is unclear.
Line 52: The expression “activates” should be changed, e. g. “enables” would be better.
Lines 72-73; 128: The Sterne vaccine is usually administered subcutaneoulsly. Are there any data available for intranasal vaccination (or other routes, e.g. oral) which could be compared to the antibody responses shown here?
Lines 125, 375; Chapter 2.3: The term “challenge” should be avoided. The swine were vaccinated with a well-known attenuated strain. They were not challenged with a virulent strain after vaccination.
Line 174/Figure 1/Table 3: I understand that the authors assumed (correctly) that a single intranasal exposure with 104 spores would result in a very weak antibody response and low bacterial shedding, and therefore samples were only taken until 28 DPI. However, what was the reason for using the same approach for animals which were inoculated once with 107 spores? The high antibody response and the presence of bacteria in 2/3 animals at 28 DPI would have justified to take samples also at 42 and 56 DPI.
Figure 1: Only the average antibody response is shown. Standard error values should be indicated by error bars. It is difficult to distinguish the colors for 104/3 exposures and 107/3 exposures.
Line 182: The wording implies that DNA was extracted directly from the nasal swabs. It would be better to write “... pending cultivation, DNA extraction from suspicious colonies and confirmation by PCR.”
Lines 215-217: The first sentence of Chapter 2.7 should be transferred to the Results part.
